# RESIDUAL ENERGY-BASED MODELS FOR TEXT GENERATION

**Yuntian Deng**[1]**, Anton Bakhtin**[2]**, Myle Ott**[2]**, Arthur Szlam**[2]**, Marc'Aurelio Ranzato**[2]
Harvard University[1]
Facebook AI Research[2]
`dengyuntian@seas.harvard.edu` `{yolo,myleott,aszlam,ranzato}@fb.com`

## ABSTRACT

Text generation is ubiquitous in many NLP tasks, from summarization, to dialogue and machine translation. The dominant parametric approach is based on locally normalized models which predict one word at a time. While these work remarkably well, they are plagued by exposure bias due to the greedy nature of the generation process. In this work, we investigate *un-normalized* energy-based models (EBMs) which operate not at the token but at the sequence level. In order to make training tractable, we first work in the residual of a pretrained locally normalized language model and second we train using noise contrastive estimation. Furthermore, since the EBM works at the sequence level, we can leverage pretrained bi-directional contextual representations, such as BERT and RoBERTa. Our experiments on two large language modeling datasets show that residual EBMs yield lower perplexity compared to locally normalized baselines. Moreover, generation via importance sampling is very efficient and of higher quality than the baseline models according to human evaluation.

## 1 INTRODUCTION

The dominant approach to parametric text generation is based on large neural auto-regressive models (Radford et al., 2019). These models can be trained efficiently via maximum likelihood and they can efficiently generate samples of remarkable quality. Key to their success is *local normalization*, i.e. they are defined in terms of a product of conditional distributions, one for each token in the sequence. Such distributions are relatively cheap to compute with modern hardware given the limited vocabulary size of common sub-word units like BPE (Sennrich et al., 2015).

Unfortunately, local normalization also brings some drawbacks. First, the designer of the model needs to specify the order in which tokens are generated. Second, at training time the model is conditioned on ground truth context while at test time it is conditioned on its own generations, a discrepancy referred to as exposure bias (Ranzato et al., 2016). Finally, while heuristics like beam search somewhat help rescore at the sequence level, generation generally lacks long-range coherency because it is produced by the greedy selection of one token at a time without lookahead.

Energy-based models (EBMs) (Hinton, 2002; LeCun et al., 2006; Ranzato et al., 2007) are a more general framework which potentially address all these issues, as they do not require any local normalization. They only require the definition of an energy function defined over the whole input sequence. Training aims at shaping the energy function such that regions of high density of training data points have lower energy than elsewhere. In principle, EBMs are ideal for modeling text as they can score the whole input at once, they are not prone to label bias (Bottou, 1991) and they may enable generation of large chunks of text, which should help improve coherency.

However, so far EBMs had limited application in text generation, because sampling from the model is intractable, and so is maximum likelihood training. The problem is that shaping the energy function is accomplished by updating the model parameters such that the energy is decreased at the training data points (a.k.a. positive examples) and increased at other data points (a.k.a. negative examples). In maximum likelihood training negatives are generated from the model, but in text application we cannot use gradient-based MCMC methods (Teh et al., 2003; Du & Mordatch, 2019) and Gibbs sampling (Welling et al., 2005) is too slow to be practical. Generating negatives by local

perturbations of the ground truth would be efficient but hardly useful for generation purposes, when at test time the model needs to generate from scratch.

Recently, Bakhtin et al. (2019) carefully studied the problem of training a discriminator to distinguish human written text from language model generations. They experimented with different language model and discriminator architectures, training/test time corpora and concluded that the discriminator can generalize rather well to weaker language models when the training/test corpora match. Bakhtin et al. (2019) found that the learned discriminator is not robust to random perturbations, and argued that the discriminator operates in the "residual" space of the language model.

Concurrently, Grover et al. (2019) proposed a general approach to "de-bias" a generator, by simply training a discriminator and using its output for importance sampling.

In this work, we build upon these two works. First, we formalize the residual interpretation by Bakhtin et al. (2019) and use a generative model of the form:

$$P_\theta(x) \propto P_{LM}(x) \exp(-E_\theta(x)) \tag{1}$$

where $P_{LM}(x)$ is a locally normalized language model which is fixed during training, and $E_\theta$ is the energy function parameterized by $\theta$. The resulting model $P_\theta(x)$ is globally normalized due to the energy term. Note that the same residual formulation was also used in Rosenfeld et al. (2001); Wang & Ou (2018b); Parshakova et al. (2019).

This formulation has multi-fold benefits. First, by incorporating a locally normalized language model, we can leverage recent advancements in locally normalized language modeling. Second, the language model provides a natural proposal distribution for training (Bakhtin et al., 2019), and training can be made efficient by using the conditional noise contrastive estimation objective (Gutmann & Hyvärinen, 2010) as we shall see in §3. Lastly, this formulation enables efficient evaluation and generation via importance sampling (Horvitz & Thompson, 1952; Grover et al., 2019).

In some sense, this last point is perhaps the central contribution of the paper, as it allows estimating perplexity of the residual EBM, and thus allows these EBMs to be compared in a standard way to other models. Indeed, in §4 we show that our joint model decreases perplexity on two large datasets, when compared to various auto-regressive language model baselines. Finally, the EBM generations are significantly preferred by humans according to our qualitative evaluation. To the best of our knowledge, this is the first time that an EBM has demonstrated improved generation ability against very strong auto-regressive baselines, both in terms of estimated perplexity and through human evaluation.

## 2 RELATED WORK

Energy-based models have a long history in machine learning (Hopfield, 1982; Hinton, 2002; LeCun et al., 2006; Ranzato et al., 2007). The key challenge of training is mining for good negatives. This can be accomplished explicitly by fantasizing inputs where the energy should be increased or implicitly via global constraints such as sparsity (Ranzato et al., 2007). Methods attempting at maximizing the likelihood of the data require to sample from the distribution induced by the model. Unfortunately, gradient-based MCMC approaches like Hybrid Monte Carlo (Teh et al., 2003) and Langevyn dynamics (Ranzato et al., 2007; Du & Mordatch, 2019; Xie et al., 2016; 2017; 2019; 2018; Gao et al., 2018; Nijkamp et al., 2019) are not applicable when the input is discrete like in text applications. Other approaches like Gibbs sampling (Hinton, 2002) were applied to binary inputs but do not scale well to large dictionaries once the energy function is a large bidirectional transformer model like the one used in this work. Several variants of auto-encoders have also been investigated for representing and generating text (Bowman et al., 2016; Zhao et al., 2018), but they have not shown significant improvements in terms of perplexity and they have so far been applied to relatively small datasets only.

Our approach appears similar to discriminative reranking approaches used in the parsing and machine translation community (Shen et al., 2004). However, our approach provides a generative model, and parameters/hyper-parameters are directly tuned to close the gap between the model distribution and the data distribution, rather than relying on surrogate ranking losses. This approach is also related to other sequence level training objectives (Edunov et al., 2018), with the major differ-

ence that in those works training aims at improving the baseline model, but generation at test time is still greedy.

Energy Networks have been used for sequence modeling (Rosenfeld et al., 2001; Wang et al., 2015; 2017; Wang & Ou, 2017; 2018a; Parshakova et al., 2019). In particular, our residual modeling form and the training algorithm is the same as in Wang & Ou (2018b), where they used an LSTM as the generator and a CNN-LSTM as the energy function, and showed significant gains compared to LSTM baselines in speech recognition. Our work builds on these prior works and develops new lower and upper bounds for the log-probability under the joint model, which makes it possible to show that the residual EBM approach gets better perplexity. We also develop an importance weighting sampling scheme used at generation time, which is focused on conditional generation as opposed to rescoring in speech recognition (Wang & Ou, 2018b). The residual EBM formalism makes it very natural to use BERT for language modeling, and we show that empirically this type of approach can outperform modern state-of-the-art language modeling baselines, both in terms of perplexity, and through human evaluation.

Generative Adversarial Networks (Goodfellow et al., 2014) also relate to EBMs, except that in EBMs the generator is implicit and negatives samples are produced by the discriminator itself. In our work, the pretrained locally normalized language model can be seen as a fixed generator, like in Bakhtin et al. (2019). Azadi et al. (2018) also share our same goal but their generator is not locally normalized and they propose to improve the sampling from the generator by using the discriminator for rejection sampling. Similar to our work, Grover et al. (2019) propose to use the discriminator to de-bias the pretrained generator using importance sampling. We adapt this work to the application of text generation. In particular, we adopt the conditional noise contrastive estimation (NCE) objective (Ma & Collins, 2018; Gutmann & Hyvärinen, 2010) to our residual model energy function and then sample from the joint model using importance sampling. We want to note that the same formulation has been proposed in (Wang & Ou, 2018b; Parshakova et al., 2019). While Ma & Collins (2018) used conditional NCE to predict the next word in a sequence, we apply it to produce a whole sequence at once with the pretrained auto-regressive language model as the noise distribution.

## 3 RESIDUAL ENERGY-BASED MODELS

We study the problem of conditional generation of discrete sequences. Given a prefix $x_1, \cdots, x_p$ with $x_j \in V$ where $V$ is the vocabulary, we want to model the probabilities of generating a sequence of total length $T > p$[1]. The generative model is:

$$P_\theta(x_{p+1}, \cdots, x_T | x_1, \cdots, x_p) = \frac{P_{LM}(x_{p+1}, \cdots, x_T | x_1, \cdots, x_p) \exp(-E_\theta(x_1, \cdots, x_T))}{Z_\theta(x_1, \cdots, x_p)} \quad (2)$$

where $Z_\theta(x_1, \cdots, x_p)$ is a normalizing factor known as partition function. Computing the partition function is intractable in our case since it involves a sum over $|V|^{T-p}$ terms which grow exponentially with the sequence length: in our experiments the size of the vocabulary is 50,096 and the length of the generation is 40 tokens. We call $P_\theta$ the joint model, and $E_\theta$ the residual energy function since $P_{LM}$ is fixed throughout training. The goal of training is to learn the parameters of the energy function such that the joint model distribution gets close to the data distribution. For the sake of reducing clutter in the notation, we will drop the conditioning variables in the following discussion.

### 3.1 TRAINING

When the partition function is intractable, Maximum Likelihood Estimation (MLE) requires samples from the model distribution, which is usually approximated with Monte Carlo sampling or mean field inference (Hinton, 2012; LeCun et al., 2006) for globally normalized models. Unfortunately, both approaches are too computationally expensive for text applications when using large bidirectional transformer models. For instance, if we were to employ Gibbs sampling exactly, we would need to perform at every position as many forward passes as words in the dictionary to compute each conditional distribution. On large datasets where training locally normalized models on multiple machines already takes days, having such additional overhead means that the model would learn

---

[1]We assume a fixed $T$ for simplicity of analysis and implementation, but our method generalizes to varying length generation with an end-of-sequence symbol.

from much less data for the same amount of time, and this is seldom a beneficial strategy for learning models that generalize well. Therefore, we do not use either MCMC nor mean field methods, as the latter would introduce additional variational parameters or an inference network which anyway yields an approximation to MLE learning.

Instead, we train our residual energy function using Noise Contrastive Estimation (NCE) (Gutmann & Hyvärinen, 2010), and more specifically its conditional version (Ma & Collins, 2018). NCE requires two distributions: The model distribution and a noise distribution. In our case, the model distribution is the joint model of Eq. 2, $P_\theta$, while the noise distribution is the pretrained language model, $P_{LM}$. NCE then trains a binary classifier on the difference of log-probability scores of these two models. Since our joint model is the product of the energy function (whose parameters we want to learn) with $P_{LM}$, the difference reduces to: $\log P_\theta - \log P_{LM} = -E_\theta$. Therefore, under these modeling assumptions of residual learning and noise model, the objective function becomes:

$$\max \mathbb{E}_{x_+ \sim P_{data}} \log \frac{1}{1 + \exp(E_\theta(x_+))} + \mathbb{E}_{x_- \sim P_{LM}} \log \frac{1}{1 + \exp(-E_\theta(x_-))} \tag{3}$$

where $x_+$ is a positive sequence taken from the human generated training set, and $x_-$ is a negative sequence drawn from $P_{LM}$ (for a given ground truth prefix). In other words, training the energy function reduces to training a binary classifier to discriminate between real text and text generated by an auto-regressive language model. The aim of training is to assign as negative energy as possible to real data, and as positive energy as possible to machine generated data. Interestingly, the role of positive and negative samples is totally symmetric in this loss function, §5 will discuss the consequences of this.

With the theoretical guarantee of NCE, we can show that the optimum of the above objective is reached at data distribution with infinite amount of data and model with enough capacity, which is also proved in Ma & Collins (2018)[2].

**Theorem 1.** *If $P_{LM}$ has the same support as $P_{data}$, then the objective function in Eq. 3 reaches its maximum at $\log P_{LM}(x) - E_\theta(x) = \log P_{data}$, if there exists such $\theta$.*

*Proof.* This theorem directly follows from the proof in Gutmann & Hyvärinen (2010). Note that at optimum, $P_{LM}(x) \exp(-E_\theta(x))$ is self-normalizing: instead of $P_\theta(x) \propto P_{LM}(x) \exp(-E_\theta(x))$, we have $P_\theta(x) = P_{LM}(x) \exp(-E_\theta(x))$. However, we still need to estimate the partition function throughout the rest of this paper, since we cannot guarantee that this optimum can be reached. □

### 3.2 EVALUATION

A commonly used protocol for evaluating generative sequence models, especially language models, is perplexity (PPL), which is equal to $2^{-\frac{1}{T-p} \sum_{i=p+1}^{T} \log_2 P(x_i | x_{i-1}, \cdots, x_1)}$. PPL can be interpreted as the average number of tokens the model is uncertain of at every time step. Since the log-likelihood required by PPL relies on estimating the partition function $Z_\theta = \sum_x P_{LM}(x) \exp(-E_\theta(x)) = \mathbb{E}_{x \sim P_{LM}} \exp(-E_\theta(x))$, we derive two estimators for the log-partition function $\log Z_\theta$ based on the work of Nowozin (2018).

**Theorem 2.** *Denote $T_n$ as the empirical estimate of $\log \mathbb{E}_{x \sim P_{LM}} \exp(-E(x))$ with $n$ samples $x_i \sim P_{LM}(i = 1, \cdots, n)$: $T_n = \log \frac{1}{n} \sum_{i=1}^{n} \exp(-E(x_i))$, then $\forall \epsilon > 0, \exists N > 0$ such that $\forall n > N$ we have*

$$Z_\theta - \epsilon < \mathbb{E}[T_n] < Z_\theta < \mathbb{E}[(2n-1)T_n - 2(n-1)T_{n-1}] < Z_\theta + \epsilon \tag{4}$$

The proof is given in Appendix A.2.

We can use the above two estimators to estimate the lower and upper bounds of the partition function, but we want to emphasize that they are true only asymptotically (when $n$ is sufficiently large). We also want to note that to get lower variance estimates we use leave-one-out strategy to estimate $T_{n-1}$. See Nowozin (2018) for implementation details and methods to improve numeric stability.

Similarly to locally normalized models, we can also factorize the probabilities of an entire sequence step by step, as $P(x) = \prod_{t=1}^{T} P(x_t | x_{<t})$, and evaluate the PPL for each generation step. By

---

[2]From Ma & Collins (2018) Assumption 2, for conditional NCE the model needs to be flexible enough such that the self-normalizing property can be satisfied conditioned on any prefix.

---

**Algorithm 1:** Top-k Joint Sampling

---

**Input:** number of samples $n$ drawn from $P_{LM}$, value of $k$ in top-k

`// Get a set of samples from P_LM`

sample $n$ samples $\{x^1, \cdots, x^n\}$ from $P_{LM}$ with top-k sampling

calculate energies $s^i = E_\theta(x^i)$ for each $x^i \in \{x^1, \cdots, x^n\}$

`// Resample from the set of LM samples`

sample $x = x^i$ with probability $\frac{\exp(-s^i)}{\sum_{j=1}^n \exp(-s^j)}$

**return** $x$

---

marginalizing over the future, we can derive the following per step probabilities:

$$P(x_t|x_{<t}) = P_{LM}(x_t|x_{<t}) \frac{\mathbb{E}_{x'_{t+1}, \cdots, x'_T \sim P_{LM}(\cdot|x_{\leq t})}[\exp(-E_\theta(x_{\leq t}, x'_{t+1}, \cdots, x'_T))]}{\mathbb{E}_{x'_t, \cdots, x'_T \sim P_{LM}(\cdot|x_{\leq t-1})}[\exp(-E_\theta(x_{\leq t-1}, x'_t, \cdots, x'_T))]}. \quad (5)$$

The step-wise probabilities in Eq. 5 are an instance of importance sampling (Horvitz & Thompson, 1952). The basic $P_{LM}$ distribution is adjusted by the probability assigned to token $x_t$ by the energy function (numerator is clamped at $x_t$ while denominator sums over all the possible values of the token at position t), with the additional marginalization over all subsequent tokens up to the horizon $T$. Since the summation involves exponentially many terms, unless $t = T$, this is approximated by samples drawn by $P_{LM}$. Since both the numerator and the denominator take the same form as the partition function, we also use Eq. 4 to estimate the upper and lower bounds. E.g., the lower bound of $\log P(x_t|x_{<t})$ can be obtained by using the lower bound of the numerator and the upper bound of the denominator.

For $t = T$, we can calculate the log probability by exhaustive enumeration. This gives us an idea of the true performance of our model at the last step, and it also provides a sanity-check of the tightness of our estimators.

### 3.3 Generation

Generating from the joint model is a non-trivial task. A naive way is to generate from the joint model auto-regressively, by marginalizing the future as in Eq. 5, which we term **Top-k auto-regressive sampling**. However, doing so is computationally expensive and impractical, and we only use this method for a qualitative analysis of the joint model in Appendix A.1.

In order to generate efficiently, we use self-normalizing importance sampling (Owen, 2013; Grover et al., 2019). Under the assumptions that the model from which we wish to draw samples is the joint model, which is the product of the auto-regressive model and the energy function, and that the proposal distribution is the auto-regressive model itself, sampling proceeds simply by: a) sampling from the auto-regressive language model, followed by b) resampling according to the energy function. The algorithm is shown in Algorithm 1, where we introduce an optional top-k constraint on the pretrained language model to improve the quality of samples in the set[3]. Without the top-k constraint, as the number of samples goes to infinity, we would recover exact samples from the joint model distribution.

## 4 Experiments

In this section, we describe the experimental set up and the results we obtained by using the residual EBM for text generation, both in terms of perplexity and generation quality.

### 4.1 Experimental Setup

**Datasets** We consider two datasets: the Toronto Book Corpus (Zhu et al., 2015; Kiros et al., 2015) and CC-News (Bakhtin et al., 2019). The former dataset consists of fiction books in 16 different

---

[3]Adapting to other types of local constraints such as nucleus sampling (Holtzman et al., 2019) is straightforward.

genres, totaling about half a billion words. The latter is a de-duplicated subset of the English portion of the CommonCrawl news dataset (Nagel, 2016), which totals around 16 Billion words. The book corpus is more challenging because the range of style and topics is more diverse than CC-News. Also, the book corpus is 30 times smaller than CC-News and may pose generalization challenges because of its smaller size.

In all our experiments we use a prefix of size 120 tokens and we generate the following 40 tokens; with the notation of Eq. 2, $p = 120$ and $T = 160$. For training the joint models, for efficiency we generated 16/128 samples per prefix for CC-News/Book Corpus offline, and sample uniformly from those samples at training time.

**Baselines**   We consider as base language model (BASE LM) used to generate negatives for the residual EBM, a transformer language model with 12 layers, $h = 16$, $d_{model} = 1024$, $d_{ff} = 4096$ (we refer to Vaswani et al. (2017) for notations). This is also our first baseline model.

The *joint* model has as many parameters as the sum of the number of parameters in the base LM and the number of parameters in the energy network. To make a fair comparison, we consider two additional baselines that have the same number of parameters as our joint model.

The first baseline is a Residual Auto-regressive Language Model baseline (RALM):

$$\log P_{RALM}(x_t|x_{<t}) = \log P_{LM}(x_t|x_{<t}) + \log P_\phi(x_t|x_{<t}) + const \qquad (6)$$

where $P_\phi$ takes the form of another auto-regressive language model. The parameters of $P_\phi$ are trained by exact maximum likelihood training of $P_{RALM}$.

The second baseline is an auto-regressive language model of the same size of our joint model (sum of the base LM and energy function parameters), we dub this model Big Auto-regressive Language Model (BALM). BALM has 12 layers, $h = 16$, $d_{model} = 1568$, $d_{ff} = 6272$, and is trained by standard token level cross-entropy loss.

**Residual EBM Architecture**   We consider two architectures for our residual EBM, both of them are based on transformers (Vaswani et al., 2017; Devlin et al., 2018). The first version uses causal self-attention and is derived from the base LM, a unidirectional transformer (UNIT). It is of the same architecture as BASE LM, except that in the final layer we project the mean-pooled hidden states to a scalar energy value. We initialize its parameters with a language model trained on the same dataset.

The second version is instead bi-directional (BIT), and the energy function is computed by projecting the mean-pooled top hidden states down to a single scalar value. We consider three variants, a BIT-BASE following the architecture of RoBERTa-Base, and a BIT-LARGE∗ following RoBERTa-Large (Liu et al., 2019), and a BIT-MED with the same number of parameters as UNIT (such that JOINT BIT-MED has roughly the same number of parameters as BALM)[4]. We initialize the parameters with a trained BERT, and we use ∗ to mark usage of external data, otherwise it means that BERT was trained on our training set. Notice how our model can be interpreted as a natural way to finetune large bidirectional pretrained models for the text generation task.

While we expect BIT to yield better results because it can fully leverage context also for intermediate tokens, we also consider UNIT to compare to the RALM baseline, which uses the same architecture and only differs in the way parameters are trained and in the presence of local normalization.

We train our models on 8 DGX nodes, each with 8 Nvidia V100s. To improve training speed, we use mixed precision training[5]. We use the Adam optimizer, with cosine learning rate decay and learning rate warmup. To stabilize training we used gradient norm clipping (Pascanu et al., 2013). Detailed hyper-parameter settings can be found in Appendix A.3.

For generation, we use top-k sampling with $k = 10$ for all human evaluations. We take 10,000 samples from BASE LM for our joint sampling.

## 4.2   RESULTS

---

[4]We use models from the HuggingFace repository at `https://github.com/huggingface/transformers`

[5]`https://github.com/NVIDIA/apex`

| Model (#parameters) | CC-News | | Toronto Book Corpus | |
| --- | --- | --- | --- | --- |
| | Val | Test | Val | Test |
| BASE LM (203M) | 18.41 | 17.57 | 16.16 | 18.29 |
| RALM (LM+203M) | 17.01 | 16.17 | 15.71 | 17.85 |
| BALM (408M) | 16.50 | 15.74 | 15.00 | 16.99 |
| JOINT UNIT (LM+203M) | 16.42-16.44 | 15.57-15.58 | 15.12-15.13 | 16.98-17.00 |
| JOINT BIT-BASE (LM+125M) | 15.32-15.35 | 14.61-14.64 | - | - |
| JOINT BIT-BASE* (LM+125M) | 15.11-15.17 | 14.37-14.42 | 14.14-14.16 | 15.72-15.74 |
| JOINT BIT-LARGE* (LM+355M) | **14.59-14.61** | **13.97-14.00** | **13.80-13.83** | **15.33-15.36** |
| BASE LM-24L (203M) | 15.71 | 14.89 | 15.61 | 18.14 |
| RALM-24L (LM-24L+203M) | 15.70 | 14.89 | 15.63 | 18.17 |
| BALM-24L (408M) | 14.58 | 13.92 | 15.20 | 18.24 |
| JOINT UNIT (LM-24L+203M) | 14.59-14.61 | 13.81-13.82 | $15.12 - 15.16$ | 17.46-17.48 |
| JOINT BIT-BASE (LM-24L+125M) | 13.68-13.69 | 13.01-13.03 | - | - |
| JOINT BIT-BASE* (LM-24L+125M) | 13.60-13.62 | 12.93-12.95 | 14.11-14.12 | 16.17-16.18 |
| JOINT BIT-MED (LM-24L+203M) | 12.97-13.01 | 12.38-12.42 | - | - |
| JOINT BIT-LARGE* (LM-24L+355M) | **12.71-12.77** | **12.10-12.16** | **13.30-13.34** | **15.17-15.22** |

Table 1: Validation and test perplexity on CC-News and Toronto Book Corpus. * denotes models initialized with RoBERTa trained on additional data. The joint model perplexity ranges are **estimated** using 100,000 samples, see Eq. 4. The number of parameters of each model is shown in parentheses.

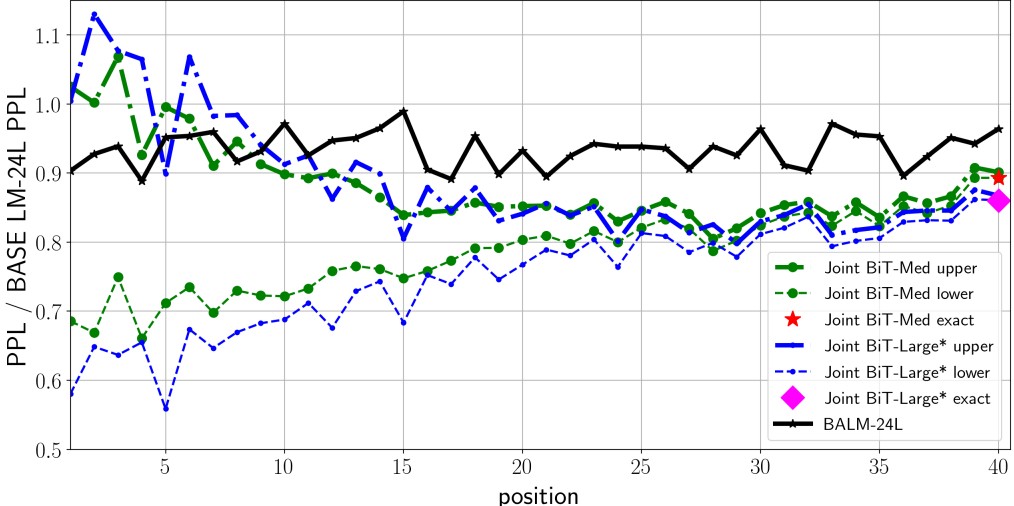

Figure 1: Perplexity gain of JOINT BIT-MED and JOINT BIT-LARGE* (using BASE LM-24L) at each position relative to BASE LM-24L on the test set of CC-News. At each position the lower and upper bounds (Eq. 5 estimated using the method in Eq. 4, see §3.2 for more details) are estimated using 20,000 samples. The shorter the horizon (moving to the right), the tighter the estimation is but also the more limited the gains compared to base LM as un-normalized models are most useful on longer generations.

**Automatic Evaluation**   Our main result is reported in Table 1 where we compare models in terms of their perplexity. We can see that on both datasets, residual EBMs with causal attention JOINT UNIT outperforms the baseline RALM with approximately the same number of parameters. The non-residual baseline BALM performs similarly to JOINT UNIT, which might be due to the limitation that $P_{LM}$ is not trained jointly with the residual model in both JOINT UNIT and RALM. However, by using our EBM approach, we can remove the causal attention mask and use bi-directional models, which achieves better performance than baselines and JOINT UNIT: without external data, JOINT BIT-BASE reaches a higher performance than JOINT UNIT with fewer parameters. By initializing from the state-of-the-art pretrained bi-directional transformers RoBERTa-Base and RoBERTa-Large, JOINT BIT-BASE* and JOINT BIT-LARGE* reach even better performance than JOINT BIT-BASE.

| Model1 (baseline) | | Model2 (compared model) | Rate | p-value |
|---|---|---|---|---|
| BASE LM | | JOINT UNIT | 52.85% | 0.16 |
| BASE LM | | JOINT BIT-BASE | 56.25% | 0.015 |
| BASE LM | | JOINT BIT-LARGE* | 58.93% | 0.00084 |
| BASE LM | | BALM | 46.77% | 0.88 |
| BALM | < | JOINT UNIT | 50.00% | 0.52 |
| BALM | | JOINT BIT-BASE | 57.89% | 0.0027 |
| BALM | | JOINT BIT-LARGE* | 59.89% | 0.00020 |
| BALM-24L | | JOINT BIT-MED (24L) | 56.23% | 0.015 |
| JOINT BIT-LARGE* (24L) | | HUMAN | 55.21% | 0.036 |
| BASE LM | ≤ | BALM | 54.85% | 0.050 |

Table 2: Human evaluation results on a subset of 333 sentences on the CC-News test set. The rate is computed as the percentage of sentences where the number of turkers preferring Model1 is strictly less than (denoted with <) or not greater than (denoted with ≤) those preferring Model2. Attention check is used to drop some votes, so there might exist ties. p-value is based on single-sided binomial test.

In the lower part of the table, we show that if we make the big language model baseline BALM deeper (BALM-24L) (24 layers instead of 12, for the same number of parameters) we attain lower perplexity. However, training the joint model JOINT BIT-BASE on the residual of a deeper language model BASE LM-24L yields even lower perplexity, despite having fewer parameters. By using the same number of parameters as BALM-24L, JOINT BIT-MED further decreases perplexity. Finally, by initializing from RoBERTa-Large, JOINT BIT-BASE* obtains the best results.

One caveat of our evaluation protocol is that the perplexity bounds are only estimates, which might not reflect the true value, particularly since the number of possible sequences grows exponentially with the number of words that are generated. We therefore break down perplexity per position in the generated sequences as in Eq. 5, and compare the estimated PPLs to the true enumerated PPLs at the last position, as shown in Figure 1. We find that at the final generation step, the estimated bounds agree remarkably well with the exact values, proving that our method at least gets a reasonable PPL estimate at the last generation step, and that JOINT BIT-MED outperforms baselines at the last generation step for sure.

**Human Evaluation**  Better perplexity results do not necessarily imply better generations. Besides, since generation from the residual EBM requires approximations as in Algorithm 1, the limited sample size might induce approximation errors compared to truly sampling from the joint distribution. Therefore, we conducted human evaluations to compare generations from the residual EBM model to generations from the baseline language models.

For each prefix, we present one completion from each model, and ask humans to select the one that is a better continuation. More details about human evaluation can be found in the Appendix A.4. The preference rates reported in Table 2 confirm that indeed the generation quality of JOINT BIT-BASE and JOINT BIT-LARGE* is better than both language model baselines. Depending on the model variant, our joint model (with bidirectional EBM) is preferred between 56% and almost 60% of the times; interestingly, the preference rate does not change much as we compare against base LM as opposed to BALM. In fact, humans do not seem to have a strong preference for BALM over base LM, despite the former scores two perplexity points lower. Similarly, JOINT UNIT is not strongly preferred over BASE LM despite its lower perplexity score. We surmise that unidirectional scoring functions and auto-regressive models exhibit generation artifacts which are easily detected by humans, and these may overshadow the improvements brought by perplexity gains.

## 4.3 ANALYSES

In this section, we analyze some of the results we obtained. First, we check whether we used a sufficient number of samples in our perplexity estimates. Second, we assess whether the joint model produces fewer repetitions compared to the base language model, and finally we check how well some statistics of the model and data distributions match.

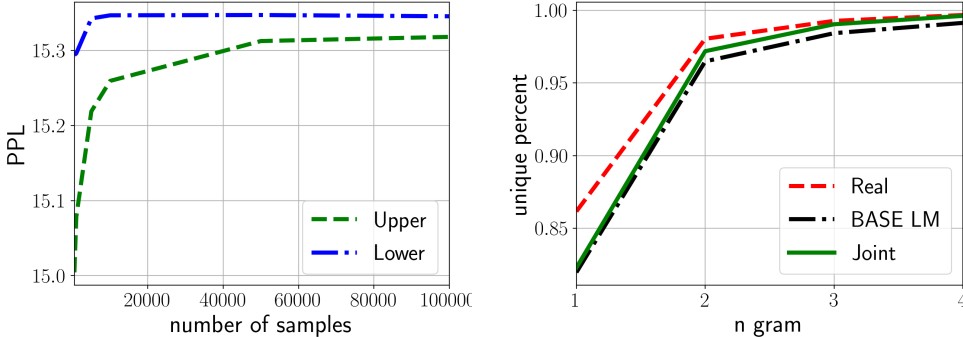

Figure 2: Left: PPL estimation for joint BIT-BASE on CC-News validation set as we vary the number of samples. Right: Percentage of Unique n-grams found in real data, samples from the joint model BIT-BASE and samples from the base language model. The joint sampling is done with 10,000 samples.

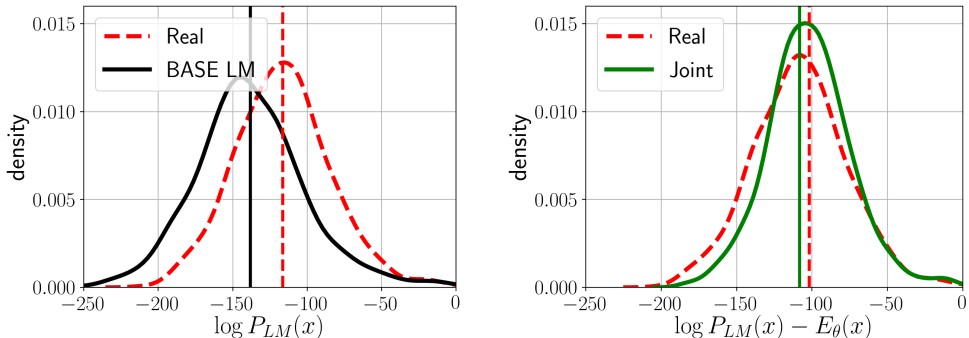

Figure 3: Density plot of log-probability scores using the base language model (left) or the joint model (right). The red curve corresponds to real samples, the black curve to samples from BASE LM and the green curve to samples from BIT-BASE. The joint model provides a much better fit than the base language model.

**Number of samples.** In Figure 2, we vary the number of samples we take in order to estimate PPL upper and lower bounds. Beyond 20,000 samples the upper estimate becomes very stable, although we have to emphasize that these estimates might be biased even though the gap between lower and upper bound closes as we take more samples.

**Repetitions.** A typical artifact of auto-regressive language models is their tendency to repeat phrases. It is then interesting to check whether the joint model is able to alleviate this artifact. Fig. 2 shows that indeed the joint model has a slightly higher percentage of unique n-grams compared to the baseline language model with $n = 2, 3, 4$, although still not as high as the original human generated text.

**A necessary condition for the model to match the data distribution.** If the joint model $p_\theta$ matches the data distribution $p_d$, then statistics computed on a large population of samples from the two distributions should also match. In particular, Fig. 3 show the density plots of log-likelihood scores of the baseline language model (left) and joint model (right) when fed with their own samples versus samples from the test set. We observe that the histogram of samples from the joint model matches the real data distribution more closely: The difference of means in the LM BASE case is 21.64 whereas the difference is 6.20 in the joint approach.

## 5 LIMITATIONS

In the previous sections we highlighted the strengths of residual EBMs, namely their simplicity, efficiency both at training and test time, and their improved perplexity scores against strong auto-regressive language model baselines. In this section, we comment on their limitations to caution the

reader about when these methods are more likely to succeed and to inform other researchers about what future avenues of research may naturally derive from this work.

In order to make training efficient and side step costly negative mining using the energy function itself, the current approach uses negatives generated from a pretrained auto-regressive language model. Therefore, our model works as long as the base language model from which we draw samples is strong enough, and as long as the ground truth and other plausible sequences are *reachable* by the baseline language model.

If the base language model has poor quality, then generation from our joint model is going to be poor as well, as the joint model merely resamples generations from the original language model. Moreover, training is going to be trivial if the base language model is poor, because the residual energy function merely needs to detect trivial generation artifacts from the base language model. In fact, observe that the role of positive and negative samples is symmetric in the loss of Eq. 3. This means that the energy function can choose to minimize the loss by either modeling the true data or the negative samples; since the latter have much simpler structure, it is going to model the negative samples. Therefore, importance sampling amounts to mostly down-weighing the worst samples from the base language model. The consequence of this is that search with a poor base language model is going to be catastrophically inefficient, as we would need to sample an impractically large number of negatives in order to find samples that are reasonably close to the true data manifold.

To summarize, this work makes a rather strong implicit assumption on the quality of the base language model, and it is expected to work well only when this is rather strong. In our application, this assumption is met quite well in practice as large auto-regressive language models trained on large datasets have improved significantly in recent years (Radford et al., 2019). In general however, residual learning always carries liability to its base model.

## 6 CONCLUSIONS AND FUTURE WORK

We investigated an EBM trained on the residual of a pretrained autoregressive language model (Wang & Ou, 2018b; Parshakova et al., 2019). The resulting joint model scores sequences holistically, thanks to the energy function. Training is very efficient and consists of a binary classification task between positives from the training set and pregenerated negatives from the fixed language model. Generation is also very efficient as it amounts to resampling from the large set of negatives produced by the base language model. Our estimates show that the resulting model has lower perplexity than the base language model. Finally, this approach may be interpreted as a natural way to finetune a large bidirectional transformer like BERT for text generation applications.

In the future, we plan to investigate other ways to generate negatives that may strike a better trade-off between the amount of compute each negative requires and their closeness to the joint model distribution. It would also be interesting to explore other loss functions and the generation of longer pieces of text by using this model auto-regressively at the chunk level, as opposed to the token level.

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

# A APPENDIX

## A.1 TOP-K AUTO-REGRESSIVE SAMPLING

In this subsection, we factorize the joint model BIT-BASE auto-regressively, and compare its differences with BASE LM. Since even estimating the per step probabilities according to Eq. 5 is too computationally expensive, we further approximate it by only considering the top 128 words predicted by BASE LM, where we sample 10,000 completions for each of them to estimate $P(x_t|x_{<t})$. Then we take the top 10 entries and re-normalize, and compare it to the top 10 probabilities of BASE LM.

Our initial explorations suggested that the joint model tends to generate fewer repetitions. Therefore we picked a few LM samples where there are repetitions at $x_t$, and use the same context $x_{<t}$ to estimate $P(x_t|x_{<t})$ for the joint model. Some examples of $P(x_t|x_{<t})$ of BASE LM and BIT-BASE are presented in Table 3. Indeed BIT-BASE usually assigns lower probabilities to repetitions even though the top k words remain the same, which is not surprising given that the existence of repetition is a strong indicator of coming from the LM, which would lead to a higher energy value hence lower joint probability.

| Context $x_{<t}$ | Model | Rank | $x_t$ | $P(x_t\|x_{<t})$ |
|---|---|---|---|---|
| [6]... is aimed at setting common benchmarks for orderly migration practices, thereby reducing irregular flows. The Global Compact contains ten guiding principles, including that migrants cannot be settled by countries with better integration policies and a fair and sustainable development. "For the first time in our history, a legally binding and | BASE LM | 0 | binding | 0.39 |
| | | 1 | legally | 0.33 |
| | | 2 | internationally | 0.06 |
| | | 3 | comprehensive | 0.05 |
| | | 4 | transparent | 0.04 |
| | BIT-BASE | 0 | binding | 0.18 |
| | | 1 | legally | 0.17 |
| | | 2 | internationally | 0.12 |
| | | 3 | comprehensive | 0.09 |
| | | 4 | transparent | 0.08 |
| [7] ... companies that land their first-choice candidates 90-100% of the time, 24% of them have "thoroughly defined" their high performer attitudes. By contrast, only 1% of companies that struggle to land their first-choice candidates "thoroughly defined" their high performer attitudes. So it seems pretty clear that companies that land their top-choice candidates are not always as willing and | BASE LM | 0 | able | 0.66 |
| | | 1 | willing | 0.09 |
| | | 2 | eager | 0.07 |
| | | 3 | ready | 0.05 |
| | | 4 | well | 0.04 |
| | BIT-BASE | 0 | able | 0.75 |
| | | 1 | willing | 0.05 |
| | | 2 | eager | 0.05 |
| | | 3 | ready | 0.04 |
| | | 4 | well | 0.03 |
| [8]... it reveals a key skill needed to lead the Fed. "You need to know what you don't know. And you need to be willing to listen when you don't know something," said Karen Dynan, who as an assistant Treasury Secretary in Barack Obama's second administration would regularly meet Fed governors. ¡EOS¿ New Delhi Dec 5 The following are mergers under review by India's financial services and | BASE LM | 0 | banking | 0.64 |
| | | 1 | financial | 0.10 |
| | | 2 | insurance | 0.09 |
| | | 3 | technology | 0.05 |
| | | 4 | IT | 0.04 |
| | BIT-BASE | 0 | banking | 0.92 |
| | | 1 | financial | 0.06 |
| | | 2 | insurance | 0.01 |
| | | 3 | technology | 0.00 |
| | | 4 | IT | 0.00 |

Table 3: Comparison of $P(x_t|x_{<t})$ between BASE LM and BIT-BASE on a few examples. Repetitions are marked with red. Only the top 5 probabilities are shown.

---

[6]Excerpt from https://www.swissinfo.ch/eng/multinational-principles_swiss-government-gives-green-light-for-un-migration-accord/44464186.
[7]Excerpt from https://www.forbes.com/sites/markmurphy/2018/05/11/this-is-the-one-piece-of-data-that-85-of-recruiters-are-missing/#25917c765dad.
[8]Excerpt from https://www.reuters.com/article/us-usa-fed-powell/fed-nominee-powell-once-hawkish-now-champions-yellens-focus-on-jobs-idUSKBN1DS0FG

## A.2 PROOF OF THEOREM 2

**Theorem 2.** *Denote $T_n$ as the empirical estimate of $\log \mathbb{E}_{x \sim P_{LM}} \exp(-E(x))$ with $n$ samples $x_i \sim P_{LM}(i = 1, \cdots, n)$, and let $T_n = \log \frac{1}{n} \sum_{i=1}^{n} \exp(-E(x_i))$, then $\forall \epsilon > 0, \exists N > 0$ such that $\forall n > N$ we have*

$$Z_\theta - \epsilon < \mathbb{E}[T_n] < Z_\theta < \mathbb{E}[(2n-1)T_n - 2(n-1)T_{n-1}] < Z_\theta + \epsilon \tag{7}$$

*Proof.* From Nowozin (2018) Eq. 35, we can write $\mathbb{E}[T_n]$ as

$$\mathbb{E}[T_n] = Z_\theta - \frac{\mu_2}{2\mu^2}\frac{1}{n} + \frac{1}{3\mu^3}\frac{\mu_3}{n^2} - \frac{1}{4\mu^4}\left(\frac{3}{n^2}\mu_2^2 + \frac{1}{n^3}(\mu_4 - 3\mu_2^2)\right)$$
$$+ \frac{1}{5\mu^5}\left(\frac{10}{n^3}\mu_3\mu_2 + \frac{1}{n^4}(\mu_5 - 10\mu_3\mu_2)\right) + o(n^{-3}) \tag{8}$$

Where $\mu = \mathbb{E}[T_n], \mu_k = \mathbb{E}[(T_n - \mu)^k]$. Equivalently,

$$\mathbb{E}[T_n] = Z_\theta - \frac{\mu_2}{2\mu^2}\frac{1}{n} + o(n^{-1}) \tag{9}$$

Therefore, $\lim_{n \to \infty} \mathbb{E}[T_n] = Z_\theta$. So $\forall \epsilon > 0, \exists N_1 > 0$ such that when $n > N_1, \mathbb{E}[T_n] > Z_\theta - \epsilon$. On the other hand, $\lim_{n \to \infty} n(Z_\theta - \mathbb{E}[T_n]) = \lim_{n \to \infty} \frac{\mu_2}{2\mu^2} + o(1) = \frac{\mu_2}{2\mu^2} > 0$, so $\exists N_2 > 0$ such that when $n > N_2$ we have $Z_\theta > \mathbb{E}[T_n]$. Up to this point, we have proved that $Z_\theta - \epsilon < \mathbb{E}[T_n] < Z_\theta$.

For the other half part of the proof, using Eq. 8 we have

$$\mathbb{E}[T_n] = Z_\theta - \frac{\mu_2}{2\mu^2}\frac{1}{n} + \frac{c}{n^2} + o(n^{-2}) \tag{10}$$

where $c$ is a constant. Therefore, $\mathbb{E}[(2n-1)T_n - 2(n-1)T_{n-1}] = (2n-1)\mathbb{E}[T_n] - 2(n-1)\mathbb{E}[T_{n-1}] = Z_\theta + \frac{\mu_2}{2\mu^2}\frac{1}{n} + o(n^{-1})$. Therefore $\lim_{n \to \infty} \mathbb{E}[(2n-1)T_n - 2(n-1)T_{n-1}] = Z_\theta$, hence $\forall \epsilon > 0, \exists N_3 > 0$ such that $\forall n > N_3 \mathbb{E}[(2n-1)T_n - 2(n-1)T_{n-1}] < Z_\theta + \epsilon$. Furthermore, $\lim_{n \to \infty} n(\mathbb{E}[(2n-1)T_n - 2(n-1)T_{n-1}] - Z_\theta) = \lim_{n \to \infty} \frac{\mu_2}{2\mu^2} + o(1) > 0$, so $\exists N_4 > 0$ such that when $n > N_4$ we have $\mathbb{E}[(2n-1)T_n - 2(n-1)T_{n-1} > Z_\theta$.

Putting the above together, $\forall \epsilon > 0$, let $N = \max\{N_1, N_2, N_3, N_4\}$, then $\forall n > N$,

$$Z_\theta - \epsilon < \mathbb{E}[T_n] < Z_\theta < \mathbb{E}[(2n-1)T_n - 2(n-1)T_{n-1}] < Z_\theta + \epsilon$$

$\square$

## A.3 OPTIMIZATION SETTINGS

| Model | fp16 | batch size | warmup steps | max steps | max lr | max grad norm |
|---|---|---|---|---|---|---|
| BASE LM | - | 32 | 2,000 | 180,000 | 0.0001 | 10 |
| RALM | - | 64 | 2,000 | 180,000 | 0.0001 | 10 |
| BALM | - | 32 | 2,000 | 180,000 | 0.0001 | 10 |
| JOINT UNIT | + | 64 | 2,000 | 180,000 | 0.0003 | 10 |
| JOINT BIT-BASE | - | 60 | 2,000 | 90,000 | 0.00005 | 0.25 |
| JOINT BIT-BASE* | - | 60 | 2,000 | 90,000 | 0.00005 | 0.25 |
| JOINT BIT-LARGE* | + | 64 | 2,000 | 90,000 | 0.0003 | 10 |
| BASE LM-24L | - | 50 | 2,000 | 90,000 | 0.0003 | 0.25 |
| RALM-24L | - | 28 | 1,000 | 90,000 | 0.00015 | 0.25 |
| BALM-24L | - | 28 | 2,000 | 90,000 | 0.0003 | 0.25 |
| JOINT UNIT (LM-24L) | + | 64 | 2,000 | 180,000 | 0.0003 | 10 |
| JOINT BIT-BASE (LM-24L) | - | 60 | 2,000 | 90,000 | 0.00005 | 0.25 |
| JOINT BIT-BASE* (LM-24L) | - | 60 | 2,000 | 90,000 | 0.00005 | 0.25 |
| JOINT BIT-MED (LM-24L) | - | 32 | 2,000 | 90,000 | 0.00005 | 0.25 |
| JOINT BIT-LARGE* (LM-24L) | - | 20 | 2,000 | 90,000 | 0.00005 | 0.25 |

Table 4: Optimization settings. We use the same setting for CC-News and Toronto Book Corpus.

The optimization settings are presented in Table 4.

Read each of the three pairs of text below and decide which is a more reasonable **extension** of the **initial words** . Note: do not worry if one or both extensions is incomplete.

---

◯ **.... 'If you try to tinker with this without the tools that only Congress has, you are as likely to break the cloud as you are to fix it,' he said. Google, which has waged similar battles with the government, and an array of other leading tech companies are supporting Microsoft in the case. Justices Sonia Sotomayor and Ruth Bader Ginsburg suggested the wait-for-Congress approach had some appeal. 'Wouldn't it be wiser just to say, 'Let's leave things as they are—if Congress wants to regulate in this brave new world, it should** just give it up,'' **Ginsburg said, according to a summary of the opinion written for the high court's concurrence. The tech companies have a history of fighting government regulations in court, and have...**

◯ **.... 'If you try to tinker with this without the tools that only Congress has, you are as likely to break the cloud as you are to fix it,' he said. Google, which has waged similar battles with the government, and an array of other leading tech companies are supporting Microsoft in the case. Justices Sonia Sotomayor and Ruth Bader Ginsburg suggested the wait-for-Congress approach had some appeal. 'Wouldn't it be wiser just to say, 'Let's leave things as they are—if Congress wants to regulate in this brave new world, it should** be regulating in this brave new world?',' wrote Sotomayor and Bader Ginsburg. A ruling is due by the end of June. If it's approved by Congress, the court could...**

Figure 4: Screenshot of the human evaluation.

## A.4  HUMAN EVALUATION

A screenshot of the human evaluation experiments can be found in Fig 4. Every page asks for 4 comparisons, one of which we know what the ground truth answer is. We subsampled 333 sentences from the test set of CC-News, and asked 3 Amazon Mechanical turkers to vote. We consider one continuation better if it gets more votes. To check the quality of the received ratings, we performed a qualification task beforehand, where one of the continuations is real text, and we kept the top half performing turkers for further evaluation (corresponding to higher than 66.67% accuracy for discriminating real from LM samples – for a total of 26 qualified turkers). Then in the actual experiment, we use one out of every four comparisons as an attention check and drop responses if the turker did not pass the check.

## A.5 QUALITATIVE ANALYSIS

We present generation examples when our approach BASE LM outperforms baseline BALM in Table 5, and when our approach underperforms in Table 6. Here the judgment is based on human evaluation when all three turkers unanimously voted in favor of one model over the other.

| Prefix | Model | Generation |
|---|---|---|
| [9]the timing of the donation couldn't be better, and even went on to say that no one is more relevant than Chance right now. Thank you to Doug and @jewelosco for donating $1,000,000 to The New Chance Arts & Literature Fund, which will ensure more students have access to arts enrichment education! #supportcps pic.twitter.com/MXZtpscU5b — SocialWorks (@SocialWorks_Chi) November 20, 2017 "In the past, CPS hasn't really gotten a lot of money from non-profits | BASE LM | . And the fact that the money is coming from the government, it makes a big difference," he said. "We're not the biggest donor of arts education, so to |
| | BALM | , and it doesn't even have the kind of resources or funding for the arts or literary programs that we need to do the things that people want to support," said Chance. |
| | BIT-BASE-WORST | ," said Cpl. Chris Bissonnette of the Stony Brook Police Department. "So the reason is, as I think about the impact of these grants and these grants to schools |
| | BIT-BASE | to fund programming. So, I think that it's a great time for us to do something to support that," said Chance. "We want to make sure that our programs |
| | GOLD | and charitable arms and organizations because there's been a past of corrupt activity in the city of Chicago," Chance said. "And so, when it comes down to it, it |
| [10]quarter. The penalties are still somewhat of a problem but tight ends Travis Kelce and Demetrius Harris made some impressive grown-man football plays. -It was nice to see running back Kareem Hunt get in the end zone for the first time since Week 3. He must feel good to end the drought. -Kelce was visibly frustrated on the sidelines and rightfully so. The officials seem to be leaning toward Oakland with calls today. Still, Kelce should've caught that easy pass that he dropped. -Quarterback Alex Smith has delivered a couple of nice deep balls to | BASE LM | get his hands on the ball this week. He threw two touchdown passes on Thursday. He should get another touchdown as the season goes on. He's got a good chance to be one of |
| | BALM | the Chiefs and the Chiefs defense has made them pay for it. The offense has done a nice job with the ball, and they's been a factor. It's nice to have |
| | BIT-BASE-WORST | tight end Martavis Bryant to get the ball back, but the throw from Smith to Davis Webb is incomplete, which could lead to an incompletion. -The Chiefs are now 5-2 after winning |
| | BIT-BASE | help the Chiefs win this game. His completion of a deep ball to wide receiver Sammy Watkins is a nice sign. -Jalen Ramsey should be back in the lineup today. The Jaguars are going |
| | GOLD | receivers Albert Wilson and Tyreek Hill. The footwork by Wilson on the deep ball was amazing and Hill's ball-tracking skills were on point as usual. Good things happen when Smith trusts |
| [11]has an amazing ability to detoxify itself namely by the liver, kidneys, and skin," she says. Detox programs and diets do sometimes make people feel better, but that happens more often if you're eating and/or drinking things that make feel like crap (something it doesn't sound like Kardashian was doing in the first place). And if, somehow, you discover that you have elevated levels of mercury or lead in your body like Kardashian said she had, it's important to figure out why, Dr. Wider says. "Finding the source is extremely important for any heavy metal, so you | BASE LM | can't say 'no or I've never seen something of that value, so I'll try and find it again and see what happens." So don't be afraid to find a way to detox or |
| | BALM | want to get the full picture and not just have to look at the whole picture," she says. "It can be a great idea to talk to your doctor to determine what's causing your symptoms." |
| | BIT-BASE-WORST | can get rid of toxins that are found in foods like whole wheat bread, nuts like walnuts, walnuts and walnuts," she says. "If a source of the metals has elevated levels of |
| | BIT-BASE | can't say, 'Oh my God, it's a lead, but I'm fine.' But it's definitely a problem in our bodies," she says. "And if it's a problem in |
| | GOLD | can lower your exposure over time," she says. Meaning, a detox diet probably isn't going to help you as much as, say, reducing your intake of foods that are high in mercury (like |

Table 5: Example generations when BIT-BASE outperforms BALM according to human evaluation. BIT-BASE-WORST shows the LM sample with the highest energy score.

---

[9]Excerpt from https://chicago.cbslocal.com/2017/11/20/chance-the-rapper-jewel-osco/

[10]Excerpt from https://chiefswire.usatoday.com/2017/12/10/halftime-analysis-for-chiefs-vs-raiders-week-14/

[11]Excerpt from https://www.glamour.com/story/kourtney-kardashian-intense-detox

| Prefix | Model | Generation |
|---|---|---|
| [12]but as the audio only came on halfway through the incident, it did not record his initial words to the girl or her response. It was also dropped in the hallway during their interaction, so it did not record any of what happened once the two left the library. Jones brought her outside to the library's front steps, where he waited for responding officers to arrive. They did not call for an ambulance, according to the lawsuit. Rather, when the officers arrived, they told her to spit blood on the grass instead of the concrete, and to move from the steps to a bench. Some 20 minutes | BASE LM | later, she was dead, according to the lawsuit. "It's not uncommon for an officer to spit blood on an unarmed, vulnerable adult. The use of spit is illegal in a public place," |
| | BALM | later, Jones was arrested for disorderly conduct and released after posting bond, the suit said. She was arrested again on May 2. In the suit, Jones is seeking more than $1 million. A |
| | BIT-BASE-WORST | later, he walked up on the ground, where they began to yell and yell again. When Jones refused, the officers ordered him not to leave or leave any more. Jones then pulled his gun on |
| | BIT-BASE | after the officers' arrival, Jones and the girl left the library and headed to the parking lot of a nearby apartment complex. "This is the type of behavior that is most likely to cause harm to |
| | GOLD | later, J.G's mother arrived and took her daughter to Cleveland Clinic in Lakewood, where she was treated for a dislocated jaw, a head injury, and an injury to her knee. |
| [13], Bronson said. "The initiative provides a variety of supports to early childhood programs' children, families and staff. The resources provided through this partnership increase the quality of the participating programs, which benefits the community and impacts our future in such a positive way," Scott said. Visit PNCGrowUpGreat.com/donorschoose. \nHere are Varsity sports headlines for April 13, 2018. Refresh to get the latest as we add news throughout the night as we collect scores: Best of the best in track and field Our Sentinel coverage area top performers lists for girls track and field | BASE LM | at the Varsity Track & Field Invite.\nThe U.S. Army Corps of Engineers has approved $2 billion in construction work along the U.S.-Mexico boundary as a way to |
| | BALM | . Check back frequently for updates. The Sentinel also has a feature on the boys basketball and wrestling teams. Boys golf The Sentinel boys golf and wrestling teams have been one of those teams who has had some |
| | BIT-BASE-WORST | .\nLONDON, April 13 (IFR) - The following are some of the main factors expected to affect Spanish corporate bond prices on Wednesday. BAML: Spanish sovereign wealth fund PPV |
| | BIT-BASE | .\nA few weeks back, it's been rumored that the HTC Desire was going to be the company's last flagship phone, and now, a new leak has confirmed that it |
| | GOLD | and boys track and field are updated going into the Saturday district meets. The season is heating up with more district and region races coming up next week. Click these links for girls top performers and boys top |
| [14]leaders meeting November 17-18 in Papua New Guinea as potential Xi-Trump meet dates. If all else fails, Trump and Xi are also expected to meet for a bit at the G20 meeting at the end of November. On the economic calendar next week, the update on jobs and the U.S. trade deficit are the headliners on November 2. Notable earnings reports: Akamai Technologies (NASDAQ:AKAM), Mondelez International (NASDAQ:MDLZ) and Olin Corp. (NYSE:OLN) on October 29; Under Armour (NYSE: | BASE LM | UAA), General Motors (NYSE:GM) on November 4; and Procter & Gamble (NYSE:PG) for October. On the retail front, Lowe's Companies (NYSE:L |
| | BALM | UA) on October 30; CVS Health (NASDAQ:CVS) on November 27; Intel Corporation (NASDAQ:INTC) on October 28; and Verizon Communications (NYSE:V |
| | BIT-BASE-WORST | UAA) and Adidas (OTCPK:ADDYYF; OTCQX:ADDYYFGF; OLYMP), on November 30; and Qualcomm Incorporated (NASDAQ: |
| | BIT-BASE | UAA), Johnson Controls (NYSE:JCI) and Cisco Systems (NASDAQ:CSCO) on November 6.\nA woman who had to have her nose and mouth taped as punishment |
| | GOLD | UAA), eBay (NASDAQ:EBAY), General Electric (NYSE:GE), Coca-Cola (NYSE:KO), Pfizer (NYSE:PFE) and Electronic Arts (NAS |

Table 6: Example generations when BIT-BASE underperforms BALM according to human evaluation. BIT-BASE-WORST shows the LM sample with the highest energy score.

[12]Excerpt from https://www.libraryjournal.com/?detailStory=lakewood-oh-mom-sues-library-over-teens-rough-treatment

[13]Excerpt from https://www.sun-sentinel.com/community/delray-sun/fl-drf-village-academy-steam-0418-20180410-story.html

[14]Excerpt from https://seekingalpha.com/article/4215142-apple-looks-to-recharge-tech-sector

