# OpenReview forum: "Residual Energy-Based Models for Text Generation"
_ICLR.cc/2020/Conference — Accept (Poster)_

### Official Review · AnonReviewer1 · 2019-10-23
**Official Blind Review #1**

**Rating:** 6

**Review:**


Contributions:

The main contribution of this paper lies in the proposed Residual Energy-based Model (EBM) for text generation. Traditional normalized models operate at the token level with MLE training, while the proposed EBM operates at the sentence level. Therefore, BERT, or RoBERTa, can be leveraged for EBM design. The residual energy function is trained via conditional NCE, which reduces to training a binary classifier to discriminate between real text and text generated by an auto-regressive language model. After model training, text can be generated by top-k joint sampling. Experiments are conducted on two large language modeling datasets. The proposed model achieves lower perplexity, and preferred by human evaluation.

Strengths:

(1) Writing & Clarity: This paper is well written, easy to follow, and clearly presented. I enjoyed reading this paper.

(2) Novelty: The proposed model contains some novelty inside. It is framed in a residual EBM framework, though by the end, the residual energy function reduces to training a binary classifier to discriminate real and fake text. Though simple, this idea is wrapped up in a nice framework. It is also interesting to observe that this sequence-level EBM regularization can be considered as a way to fine-tune BERT for the text generation task.

(3) Experiments: Generally, the experiments are comprehensive. Detailed analysis, and human evaluation is also provided.

Weaknesses:

(1) Clarity: I have some concerns regarding the selection of baselines, with details shown below.

This paper is basically about using BERT as a binary classifier, which serves as a residual energy function to regularize a pre-trained language model and provides sequence-level supervision. The experiments are comprehensive, but on the other hand, it is also quite expected that the proposed model should work better than an MLE baseline, since sequence-level supervision is provided.

I think if the authors want to make a stronger paper, they should also compare with other possible ways to inject sequence-level supervision. For example, a simple solution is to use GAN, like in a SeqGAN setup. And the discriminator in the GAN will be the same BERT-based binary classifier. In this GAN setup, sequence-level supervision is also provided.

Then the difference is that in the GAN setup, the BERT-based binary classifier is a discriminator, but in this paper's setup, it is a residual energy function. It would be interesting to discuss and conduct experiments to see which way is better.

(2) Experiments: I have some concerns regarding the experimental setup.

a) One of the main results is Table 1, which reports all the PPL numbers. However, reporting PPL results is less interesting, because we also care about the diversity of generated samples. Lower PPL does not necessarily mean higher-quality text. Though Figure 2 provides some analysis on the diversity, a more comprehensive evaluation on this will be appreciated.

b) It will be good if the authors can also provide some generated samples for qualitative analysis.

Overall, I think this paper is well executed. The paper is well written, and experiments are carefully conducted. However, on the other hand,  I also think the conclusion in this paper is expected, it only shows that the proposed model is better than an MLE baseline.


**Experience Assessment:**

I have published one or two papers in this area.

**Review Assessment: Checking Correctness Of Derivations And Theory:**

I assessed the sensibility of the derivations and theory.

**Review Assessment: Checking Correctness Of Experiments:**

I assessed the sensibility of the experiments.

**Review Assessment: Thoroughness In Paper Reading:**

I read the paper at least twice and used my best judgement in assessing the paper.

---

> ### Author Response · Authors · 2019-11-14
> **Response to Review #1**
>
> Thank you so much for your helpful comments!
>
> - Comparison to SeqGAN:
> Compared to SeqGAN, the difference is that our goal is to use the "discriminator" (energy network) to improve the generator at test time whereas SeqGAN would throw away the discriminator after training and use the improved generator. SeqGAN requires policy gradients, which is usually unstable, and a too-powerful discriminator would make the generator gradients vanish (Arjovsky et al 2017). GANs "have proven difficult to train in the domain of natural language", "most models yield worse results than a simple Language Model", "can be extremely sensitive to the random initialization and small deviations from the best hyperparameter choice", and "extensive evaluation has shown that existing language GANs do not improve over maximum likelihood-trained models" (d’Autume et al 2019, Semeniuta et al 2018, Caccia et al 2018).
> To the best of our knowledge, there's no existing work that used a discriminator as powerful as BERT to successfully train language GANs, and in practice we observed that BERT is able to distinguish real text from LM samples over 95% of the time, hence language GAN learning with such a powerful discriminator seems implausible, not to mention that due to our large datasets and models, it is hard to do hyper-parameter tuning to stabilize policy gradients based training.
>
> - PPL:
> Perplexity is the standard metric for language modeling. It is true that other metrics could be considered, like diversity. We don't think that our approach would improve diversity compared to the original language model (our generator).
> The main reason people worry about generation diversity in GANs is because of the mode collapsing problem (e.g., a generator always generating "I don't know" might fool the discriminator). However, unlike GANs which only use the generator at test time, we use the energy function to adjust the original language model to better approximate data distribution, so we don't see mode collapsing problems in our approach. One empirical evidence is that we found the adjusted per-step probabilities are largely similar to the original language model probabilities (Appendix A.1), and empirically language models do not have as severe mode collapsing problems as GANs.
> One could improve diversity by sampling hypotheses sequentially from the joint model adding an additional constraint on diversity with respect to the previously drawn samples, or use temperatured sampling. We believe this is an interesting avenue of future research.
>
> - Qualitative analysis:
> Thank you for the suggestion. We have added some examples to the supplementary material (A.5).
>
> - Conclusions are expected:
> Residual EBMs provide a very natural way of leveraging BERT for language modeling, and we believe that providing a simple working recipe to use unnormalized sequence-level generative modeling to improve very large state-of-the-art language models is an important contribution.
>
> References:
> Arjovsky et al 2017: Towards Principled Methods for Training Generative Adversarial Networks
> d’Autume et al 2019: Training language GANs from Scratch
> Caccia et al 2018: Language GANs falling short
> Semeniuta et al 2018: On accurate evaluation of GANs for language generation

---

### Official Review · AnonReviewer3 · 2019-10-27
**Official Blind Review #3**

**Rating:** 6

**Review:**

The authors make good points, starting from the exposure bias and label bias suffered by the mainstream neural auto-regressive models.
Residual EBMs are defined and trained using NCE. Experiments on two large language modeling datasets show that residual EBMs yield lower perplexity and generation via importance sampling is of higher quality, compared to locally normalized baselines.

In generally, the paper is well motivated and interesting. But I have some concerns.

1. Missing important relevant references.

EBMs (a.k.a. un-normalized models, random fields) have been successfully developed in language modeling in recent years. A large body of this paper has been studied in [5,6], including the model and the NCE estimation method. The model Eq.(2) is exactly the model in [5], defining the model in the form of exponential tilting of a reference distribution.
Connecting and comparing to these previous works are needed.

[1] R. Rosenfeld, S. F. Chen, and X. Zhu, “Whole-sentence exponential language models: a vehicle for linguistic-statistical integration,” Computer Speech & Language,  2001.
[2] B. Wang, Z. Ou, and Z. Tan, “Trans-dimensional random fields for language modeling,” ACL, 2015.
[3] B. Wang, Z. Ou, and Z. Tan, “Learning transdimensional random fields with applications to language modeling,” IEEE transactions on pattern analysis and machine intelligence, 2018.
[4] B. Wang and Z. Ou, “Language modeling with neural trans-dimensional random fields,” IEEE Automatic Speech Recognition and Understanding Workshop (ASRU), 2017.
[5] B. Wang and Z. Ou, “Learning neural trans-dimensional random field language models with noise-contrastive estimation,” IEEE International Conference on Acoustics, Speech and Signal Processing (ICASSP), 2018.
[6] B. Wang and Z. Ou, “Improved training of neural trans-dimensional random field language models with dynamic noise-contrastive estimation,” IEEE Spoken Language Technology Workshop (SLT), 2018.

2. I am a little bit concerned that the theoretical contribution seems weak.
Though Eq. (4) and (5) seem to be novel, I am not sure whether such a contribution is substantial enough to motivate acceptance.

I'm happy to adjust the score if the paper can be better placed in the literature and the authors take efforts to improve the paper.

--------update after reading the response-----------
Being well-placed in the literature and properly claiming contribution with respect to prior work is one of the key questions in reviewing a paper. The first version of the paper clearly lacks in this respect. That's the main concern when I gave a 1.

I appreciate the authors' response. The updated paper has been improved to address my main concern, although the added discussions presented in the updated paper is not as clear as the authors' clarifications in the response. I suggest to polish the main text incorporating these clarifications.

Generally, it is nice to see the successful application of energy-based/random-field-based models in text generation, besides in speech recognition. I update the score to 6 (Weak Accept).

It would have been better that the following can be further clarified.

"the partition function estimated via importance sampling would lead to bias favoring the random field language model" --- this comment is not clear to me.

Both Eq.4 and Eq.5 give estimates for perplexity. It would be better to clarify different uses of the two equations. If the perplexities are estimated using Eq.4 (as in Table 1), then what is the purpose of developing Eq.5?

How to calculate the lower and upper bounds of the step-wise perplexity gain at each position in Figure 1?

Under Figure 1, "At each position the lower and upper bounds (see Eq. 4) are estimated using 20,000 samples." But in the main text, it is said that "We therefore break down perplexity per position in the generated sequences as in Eq. 5" at page 8. It is confusing.

**Experience Assessment:**

I have published in this field for several years.

**Review Assessment: Checking Correctness Of Derivations And Theory:**

I carefully checked the derivations and theory.

**Review Assessment: Checking Correctness Of Experiments:**

I carefully checked the experiments.

**Review Assessment: Thoroughness In Paper Reading:**

I read the paper thoroughly.

---

> ### Author Response · Authors · 2019-11-14
> **Response to Review #3**
>
> We thank the reviewer for the useful feedback, and especially for the pointers to prior work.  This paper is improved due to your care!
>
> - Missing references and unclear contribution:
> We thank the reviewer for pointing this out. The revised version of the paper has now a whole new paragraph discussing the relation to prior works on energy-based models for sequence generation [1, 2, 3, 4, 5, 6]. In particular, the residual modeling form and the training algorithm is the same as in [5], of course with different choice of generator (transformer in our case VS LSTM in [5]) and energy function (BERT in our case VS CNN-LSTM-based model in [5]). Therefore, the modeling form and loss function should not be considered our contribution.
>
> Our theoretical contributions are: a) new lower and upper bounds for the log-probability of the joint model which allows us to show that these models have better perplexity compared to autoregressive approaches (since otherwise the partition function estimated via importance sampling would lead to bias favoring the random field language model), b) the importance weighting sampling scheme used at generation time, and c) the setting which is focused on conditional generation as opposed to rescoring for speech recognition.
>
> In particular, (a) is important because it allows comparing the EBMs (with bi-directional models) against auto-regressive models in the standard metric by which these methods are judged; and we do indeed show that the residual EBMs get good results even compared against large SOTA LMs. We also show this with human evaluations.  In our opinion, improving upon these modern language models is an exciting accomplishment.
>
> Please, let us know if our revised discussion of prior work needs further clarifications. Thank you.

---

### Official Review · AnonReviewer2 · 2019-11-01
**Official Blind Review #2**

**Rating:** 6

**Review:**

This work is an interesting extension of Gutmann and Hyvarinen (2010), where the parametric model is the combination of a noise model (language model) and an energy function (residual energy), so the difference of parametric model and the noise model cancels out the noise model. Therefore optimizing (3) under some conditions converges to a set of parameters of the parametric model (P\theta(x) here) that best describes the data.

One important assumption of Gutmann and Hyvarinen (2010) is that there exists a set of optimum parameters for the parametric model such that the probability of data and the parametric models match for these optimum parameters. This should be mentioned in Theorem-1.

Does Theorem-1 need extra parameters to act as a normalization constant in order for the theorem to hold at the optimum?
log P_lm(x) - E(x) + const = log p_data

To sample from the model, the authors first sample from the language model and re-sample it with respect to the energy values of the residual model.


To compute the perplexity, they have given an upperbound and lowerboud for the partition function based on number samples in Theorem 2, but I haven't checked the correction of the bounds.  They also factorize the joint model in auto-regressive factorization to compute the perplexity by approximate marginalizing.


As mentioned in Section 5, this approach heavily depends on a strong pretrained language model.

Have you considered improving the language model during training?

The described idea is simple and effective and I really liked it.


--- Based on other reviews and the authors' response (especially review #3), I reduced my rating to 'Weak accept'.

**Experience Assessment:**

I have read many papers in this area.

**Review Assessment: Checking Correctness Of Derivations And Theory:**

I assessed the sensibility of the derivations and theory.

**Review Assessment: Checking Correctness Of Experiments:**

I carefully checked the experiments.

**Review Assessment: Thoroughness In Paper Reading:**

I read the paper thoroughly.

---

> ### Author Response · Authors · 2019-11-14
> **Response to Review #2**
>
> Thank you so much for your helpful comments!
>
> - Missing assumption on the existence of the solution:
> The reviewer is correct. The revised version of the paper now makes this assumption explicit in Theorem 1.
>
> - Partition function needs extra parameter:
> As proven in Ma & Collins and reported in the paper, the assumption is that the amount of training data goes to infinity and that the model has sufficient capacity. There is no need for additional parameters under this assumption since a powerful energy function would learn this normalizer.
> In practice, we observed that the score produced by the transformer energy function after training with the NCE objective is well calibrated as we vary the conditioning prefix. No additional bias parameter is required.
>
> - Alternating between training the residual energy function and the generator:
> This is possible. In this work, we focused on improving the original generator by only training the residual energy function which is the simplest setting. However, iterative training like in GANs might further improve the overall model.
> Since the transformer models we use in this work are big and slow to train, and training sequence GANs requires policy gradients and heavy tuning to control gradient variance, this direction requires significant engineering efforts. Alternatively, a knowledge-distillation like procedure where we use the generated samples from the joint model to fine-tune the generator might also work. We leave this as future work.

---

### Public Comment · ~Jianwen_Xie1 · 2020-01-02
**Missing related reference about un-normalized energy-based models parameterized by neural nets for image/video/3D shape generation**

Dear Authors,

Congratulations on your nice accepted paper about text generation by un-normalized energy-based models.

I would like to point out some papers that are highly related to your current one, and hope you can cite them in your final version.  Similar to your current paper, all of them are about un-normalized energy-based models parameterized by modern neural nets for image/video/3D shape generation.  The learning is based on MLE. The sampling is based on Langevin dynamics.

More specifically,

The first paper that proposes an energy-based model parameterized by modern deep neural network and learned it by Langevin based MLE is in (Xie. ICML 2016) [1].  The model is called generative ConvNet, because it can be derived from the discriminative ConvNet.  (Xie. ICML 2016) [1] originally studied such an EBM model on image generation  theoretically and practically in 2016.

(Xie. CVPR 2017) [2] (Xie. PAMI 2019) [3] proposed to use Spatial-Temporal ConvNet as the energy function for video generation.  The model is called Spatial-Temporal generative ConvNet.

(Xie. CVPR 2018) [4] proposed to use volumetric 3D ConvNet as the energy function for 3D shape pattern generation. It is called 3D descriptor Net.

(Gao. CVPR 2018) [5] proposed multi-grid MCMC to learn EBM with ConvNet as energy function for image generation.

(Nijkamp. NeurIPS 2019) [6] proposed short-run MCMC to learn EBM with ConvNet as energy function for image generation.

Thank you :)

Reference
[1] A Theory of Generative ConvNet.
Jianwen Xie *, Yang Lu *, Song-Chun Zhu, Ying Nian Wu (ICML 2016)

[2] Synthesizing Dynamic Pattern by Spatial-Temporal Generative ConvNet
Jianwen Xie, Song-Chun Zhu, Ying Nian Wu (CVPR 2017)

[3] Learning Energy-based Spatial-Temporal Generative ConvNet for Dynamic Patterns
Jianwen Xie, Song-Chun Zhu, Ying Nian Wu
IEEE Transactions on Pattern Analysis and Machine Intelligence (TPAMI) 2019

[4] Learning Descriptor Networks for 3D Shape Synthesis and Analysis
Jianwen Xie *, Zilong Zheng *, Ruiqi Gao, Wenguan Wang, Song-Chun Zhu, Ying Nian Wu (CVPR) 2018

[5]  Learning generative ConvNets via multigrid modeling and sampling.
R Gao*, Y Lu*, J Zhou, Song-Chun Zhu, Ying Nian Wu (CVPR 2018).

[6] On learning non-convergent non-persistent short-run MCMC toward energy-based model.
E Nijkamp, M Hill, Song-Chun Zhu, Ying Nian Wu (NeurIPS 2019)

---

> ### Author Response · Authors · 2020-02-15
> **References Updated**
>
> Thanks for your comments! Compared to images, a challgenge in text modeling is that the inputs are discrete, such that we cannot directly apply Langevin dynamics (our initial attempts at working in a relaxed space were not very successful). That being said, these prior works on EBMs for images/videos are indeed relevant, and we've updated our references accordingly.

---

> > ### Public Comment · ~Jianwen_Xie1 · 2020-04-30
> > **Thanks!**
> >
> > Thank you so much! Wish you all a successful and healthy 2020!

---

### Public Comment · ~Ning_Miao1 · 2020-05-15
**Hope to see comparison with some existing energy-based text generation methods, especially MCMC-based methods.**

Dear Authors,

Congratulations on the accepted paper!

In this paper, you mentioned MCMC methods for energy-based text generation. Actually, we proposed such a method called CGMH (<CGMH: Constrained Sentence Generation by Metropolis-Hastings Sampling>) in last year's AAAI.

While your proposed method generates samples by importance sampling, CGMH performs lexical-level jumps between sentences to generate samples from an unnormalized distribution (such as P_{\theta} in your paper), in the spirit of Metropolis-Hastings sampling.

CGMH is also a highly efficient energy-based text generation model, which achieves remarkable results on several constrained-text-generation tasks such as keyword-to-sentence generation and unsupervised paraphrase. Hope we will have the opportunity to discuss and compare these two models.

Thank you.

---

### Decision · Program_Chairs · 2019-12-19

**Decision:**

Accept (Poster)

**Comment:**

This paper proposes a Residual Energy-based Model for text generation.

After rebuttal and discussion, the reviewers all converged on a vote to accept, citing novelty and interestingness of the approach.

Authors are encouraged to revise to address reviewer comments.